# Triclosan depletes the membrane potential in *Pseudomonas aeruginosa* biofilms inhibiting aminoglycoside induced adaptive resistance

**Michael M. Maiden**[1,2¤], **Christopher M. Waters**[1,2]*

**1** Department of Microbiology and Molecular Genetics, Michigan State University, East Lansing, Michigan, United States of America, **2** The BEACON Center for The Study of Evolution in Action, Michigan State University, East Lansing, Michigan, United States of America

¤ Current address: Beaumont Children's Hospital, Royal Oak, MI, United States of America
* watersc3@msu.edu

**Data Availability Statement:** These genome sequence cited in this paper can be found at NCBI with BioSample accessions: SAMN15325472, SAMN15325473, SAMN15325474,

## Abstract

Biofilm-based infections are difficult to treat due to their inherent resistance to antibiotic treatment. Discovering new approaches to enhance antibiotic efficacy in biofilms would be highly significant in treating many chronic infections. Exposure to aminoglycosides induces adaptive resistance in *Pseudomonas aeruginosa* biofilms. Adaptive resistance is primarily the result of active antibiotic export by RND-type efflux pumps, which use the proton motive force as an energy source. We show that the protonophore uncoupler triclosan depletes the membrane potential of biofilm growing *P. aeruginosa*, leading to decreased activity of RND-type efflux pumps. This disruption results in increased intracellular accumulation of tobramycin and enhanced antimicrobial activity *in vitro*. In addition, we show that triclosan enhances tobramycin effectiveness *in vivo* using a mouse wound model. Combining triclosan with tobramycin is a new anti-biofilm strategy that targets bacterial energetics, increasing the susceptibility of *P. aeruginosa* biofilms to aminoglycosides.

## Author summary

Adaptive resistance is a phenotypic response that allows *P. aeruginosa* to transiently survive aminoglycosides such as tobramycin. To date, few compounds have been identified that target adaptive resistance. Here, we show the protonophore uncoupler triclosan disrupts the membrane potential of *P. aeruginosa*. The depletion of the membrane potential reduces efflux pump activity, which is essential for adaptive resistance, leading to increased tobramycin accumulation and a shorter onset of action. Our results demonstrate that in addition to its canonical mechanism inhibiting membrane biosynthesis, triclosan can exert antibacterial properties by functioning as a protonophore that targets *P. aeruginosa* energetics.

SAMN15325475, SAMN15325476,
SAMN15325477, SAMN15325478,
SAMN15325479, SAMN15325480,
SAMN15325481.

**Funding:** This work was supported by grants from
the Hunt for a Cure Foundation, the NSF BEACON
center for evolution in action (DBI-0939454), NIH
grants GM109259, GM110444, and AI143098 to C.
M.W. and the Cystic Fibrosis Foundation
Traineeship to M.M.M. In addition, M.M.M. was
supported by a Wentworth Fellowship and Rudolf
Hugh award from the MSU Department of
Microbiology and Molecular Genetics and by a
Dissertation Completion Fellowship from the
College of National Science. The funders had no
role in study design, data collection and analysis,
decision to publish, or preparation of the
manuscript.

**Competing interests:** The authors have declared
that no competing interests exist.

## Introduction

*Pseudomonas aeruginosa* is a Gram-negative opportunistic pathogen that can form highly recalcitrant biofilms in immunocompromised hosts such as those suffering from cystic fibrosis (CF) and diabetes [1–4]. Biofilms are a community of cells enmeshed in a self-made matrix that is highly tolerant to antibiotic treatment [1–4]. In addition to biofilm growth, *P. aeruginosa* possesses various layers of antibiotic resistance including low outer membrane (OM) permeability, which is estimated to be 1/100$^{th}$ that of *Escherichia coli* [5], the expression of proton motive force (PMF) dependent resistance-nodulation-cell division (RND) family of multidrug efflux pumps, and a chromosomal encoded AmpC β-lactamase, making it naturally resistant to β-lactams [4].

*P. aeruginosa* also exhibits a unique and poorly understood form of inducible antibiotic resistance, known as "adaptive resistance", in response to cationic antimicrobials such as aminoglycosides, antimicrobial peptides, and polymyxins among others [6,7]. This form of resistance is transient and triggers an induction in gene expression, protein production, or alteration in antibiotic targets [7–22]. Adaptive resistance results in alteration in the lipid A anchor of lipopolysaccharides (LPS) in response to polycationic antimicrobials due to the induction of regulatory proteins in classic two component systems like PhoP-PhoQ, PmrA-PmrB, CprR-CprS, and ParR-ParS [15–17]. Another molecular mechanism leading to adaptive resistance is the induction of RND-type efflux pumps [11,12,18,19]. In several reports, RND-type efflux pumps are the major mediator of adaptive resistance, and they can be induced within 2-hrs of exposure to subinhibitory concentrations due to diffusion limitation in biofilms or sub-therapeutic dosing [11,12,18,19, 23]. Specifically, aminoglycosides cause the mistranslation of the PA5471 leader peptide PA5471.1, a peptide which inhibits the MexZ repressor, thereby triggering the induction of the RND-type efflux pump MexXY [24]. Although adaptive resistance is not exclusive to the biofilm lifestyle, it is induced by exposure to subinhibitory concentrations of antibiotics, which can occur in biofilms due to reduced diffusion of cationic antimicrobials [7–22]. The phenomenon of adaptive resistance has been observed *in vitro*, in animal models, and in CF patients [8,14,20–22].

Despite the well-documented phenomenon of adaptive resistance, aminoglycosides continue to be the standard of care for pseudomonal infections [25]. In fact, pulmonary infections in CF patients are treated with extended doses of the aminoglycoside tobramycin (300-mg nebulized twice-a-day for 28-days), reaching mean sputum concentrations of ~737 μg/g per dose [25,26]. Aminoglycoside uptake occurs in three steps [27–29]. In the first step, termed "self-promoted uptake," positively charged aminoglycosides bind negatively charged components of the OM, triggering permeabilization and diffusion [27–29]. In the second step, energy dependent phase-I (EDP-I), aminoglycosides slowly cross the inner membrane (IM) dependent upon the membrane potential (Δψ) [27–29]. In the third step, EDP-II, aminoglycosides bind to the A-site of the 30S subunit of membrane-associated ribosomes and cause the synthesis of misfolded proteins that insert into the IM triggering permeabilization of the cytosolic barrier [27–29]. This leads to an irreversible entry of aminoglycosides into the cell, thereby inhibiting translation.

We previously performed a high throughput screen (HTS) to identify compounds that enhanced tobramycin against *P. aeruginosa* biofilms and unexpectedly discovered that the combination of triclosan and tobramycin (or other aminoglycosides) exhibited 100-times more maximum killing of established *P. aeruginosa* biofilms than either triclosan, tobramycin, or other aminoglycosides alone [30]. Triclosan is a fatty acid synthesis inhibitor, targeting the enoyl-acyl carrier reductase FabI [31]. However, it is well-known that *P. aeruginosa* is inherently resistant to triclosan due to the expression of both the RND-type efflux pump TriABC

and a triclosan resistant enoyl-acyl carrier reductase FabV, and our previous results suggested that triclosan does not enhance tobramycin activity by targeting FabI [30,32,33]. Therefore, the mechanism of action of triclosan enhancement of aminoglycoside activity remained unknown.

Recently, it has been shown that triclosan possesses protonophore activity that can disrupt the membrane potential ($\Delta\Psi$) of mitochondria isolated from rats, artificial lipid bilayer membranes, and the Gram-positive bacterium *Bacillus subtilis* [34–38]. The $\Delta\Psi$ is one component of the PMF, which is the sum of the $\Delta\Psi$ due to charge separation (positive$_{outside}$/negative$_{inside}$) and the proton gradient ($\Delta$pH, acidic$_{outside}$/alkaline$_{inside}$) across the IM [39,40]. These findings led us to hypothesize that triclosan enhances aminoglycoside activity by targeting the $\Delta\Psi$ in *P. aeruginosa* [30]. In fact, from the same HTS in which triclosan was identified, we discovered that the protonophore uncoupler oxyclozanide also acted synergistically with tobramycin to kill *P. aeruginosa* biofilms [41].

As antibacterial resistance continues to be an emerging public health crisis, there is a crucial need for new therapeutic approaches that decrease resistance [42]. One approach is to target bacterial energetics [43]. The feasibility of this approach was recently demonstrated by the discovery of the protonophore uncoupler bedaquiline, which is the first new Food and Drug Administration (FDA) approved *Mycobacterium tuberculosis* (*Mtb*) drug in 40-years [44–46].

Here we demonstrate that triclosan increases *P. aeruginosa* susceptibility to tobramycin by disrupting adaptive resistance. This occurs because triclosan decreases the $\Delta\Psi$, leading to a reduction in the activity of RND-type efflux pumps, which increases tobramycin accumulation within cells. Finally, we demonstrate that triclosan in combination with tobramycin embedded in a hydrogel is effective *in vivo* using a murine wound model. As numerous toxicity studies have demonstrated that triclosan is safe when used appropriately [47–49], this combination has the clinical potential to improve the treatment of biofilm-based infections by targeting bacterial energetics.

## Results

### The disruption of fatty acid synthesis does not increase *P. aeruginosa* susceptibility to tobramycin

To determine if the inhibition of fatty acid biosynthesis and membrane biogenesis by triclosan is responsible for increasing *P. aeruginosa* susceptibility to tobramycin, we tested if the specific fatty acid synthesis inhibitor AFN-1252 synergized with tobramycin to kill *P. aeruginosa* biofilms. AFN-1252 forms a binary complex with the active site of FabI, an enoyl-acyl reductase responsible for the final elongation step in fatty acid synthesis and the molecular target of triclosan [50]. We reasoned that if triclosan inhibition of FabI was the mechanism responsible for increased activity of tobramycin, we should similarly observe synergy of AFN-1252 and tobramycin to kill *P. aeruginosa* biofilms.

AFN-1252 alone and in combination with tobramycin was not effective at killing biofilms (S1 Fig). At the maximal concentrations of tobramycin and AFN-1252 (400 μM) used we observed slightly more than 2-fold killing; however, this difference was not statistically significant and far weaker than the ~100-fold killing observed when triclosan is combined with tobramycin [30]. These findings are in agreement with our previous genetic experiments using a FabI mutant of *P. aeruginosa* that showed identical resistance to tobramycin treatment as the parental strain [30]. Together, both our genetic and molecular evidence suggest that the inhibition of FabI by triclosan does not account for increased *P. aeruginosa* susceptibility to tobramycin.

## Triclosan combined with tobramycin causes synergistic permeabilization of cells within biofilms

Aminoglycosides corrupt protein synthesis by binding to the A-site of the 30S subunit of membrane bound ribosomes causing mistranslation of membrane proteins and inner membrane permeabilization [27–29]. We initially hypothesized that triclosan enhanced tobramycin killing by permeabilizing cells in biofilms. To test this hypothesis, we stained cells with the cell-impermeant TO-PRO™-3 iodide dye, which specifically emits a fluorescent signal only in permeabilized cells.

Treatment for 2-hrs with triclosan or tobramycin alone resulted in ~15% and ~11% of the cells within biofilms to become permeabilized, respectively (Fig 1). Neither of these increases were statistically significantly compared to untreated controls. Alternatively, the combination of triclosan and tobramycin permeabilized ~50% of cells after only 2-hrs of treatment. This result suggests that neither triclosan nor tobramycin treatment alone is sufficient to permeabilize *P. aeruginosa* cells growing in biofilms, but the combination results in increase cell permeabilization.

## The selection of mutants resistant to the combination

After eliminating inhibition of fatty acid synthesis and enhanced permeabilization as a possible explanation for the mechanism of action of triclosan [30], we turned to an unbiased experimental evolution approach to identify this mechanism [51]. The principle behind this approach is that one can select for resistance to a given drug or combination and then perform re-sequencing to identify the intracellular genetic target(s).

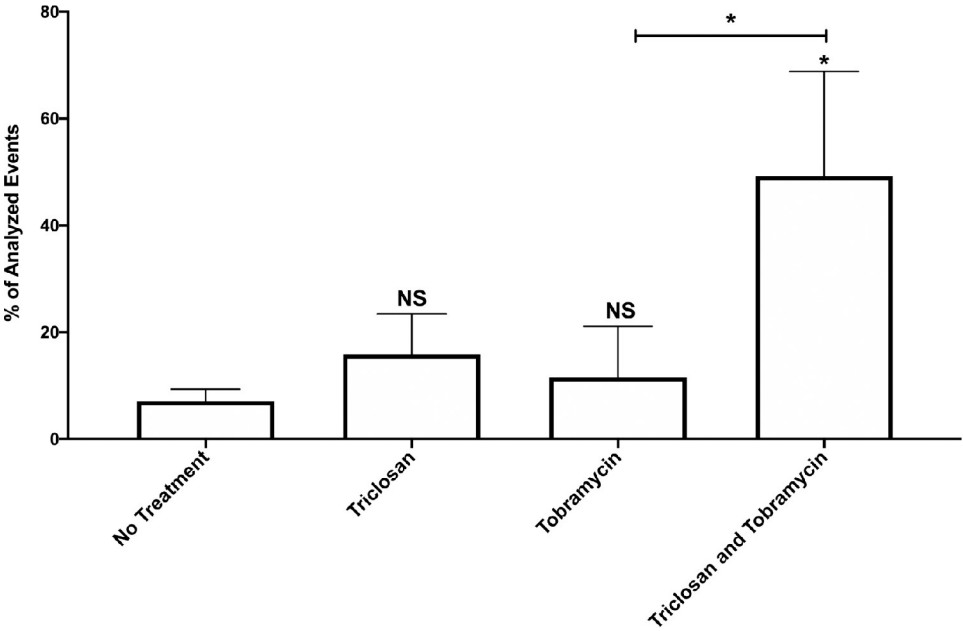

**Fig 1. Triclosan in combination with tobramycin results in increased permeabilization of cells within biofilms.** 24-hr old biofilms were treated with triclosan (100 µM), or tobramycin (500 µM), alone and in combination for 2-hrs. Cells were stained with the live/dead cell indicator TO-PRO™-3 iodide to determine the number of cells that were permeabilized. The results are percent averages plus the standard deviation (SD, n = 4). Percent values indicate the average relative abundance of events within each gate normalized to the total number of events analyzed, excluding artifacts, aggregates and debris. A one-way analysis of variance (ANOVA) followed by Dunnett's multiple comparison post-hoc test was used to determine statistical significance between each group and the untreated control and Bonferroni's post-hoc test was performed to compare tobramycin vs tobramycin and triclosan. *, p<0.05. NS, not significant.

We had previously demonstrated that triclosan does not enhance tobramycin killing of planktonic *P. aeruginosa* cells growing exponentially by simultaneous dilution of both tobramycin and triclosan from high concentrations to low concentrations [30]. We extended these experiments by examining the minimum inhibitory concentration (MIC) of tobramycin required to kill planktonic exponentially growing *P. aeruginosa* with or without 100 μM triclosan. In support of our previous conclusion, triclosan did not impact the MIC of tobramycin against planktonic cells, although it did reduce the maximal growth yield in a tobramycin-independent fashion (S2 Fig).

Because the combination does not exhibit synergistic activity against planktonic cells, we screened for resistant mutants in biofilm growing bacteria using a modified method developed by Lindsey et al. [52]. 27 *P. aeruginosa* biofilms serially passaged in parallel were exposed to sudden, moderate, and gradual treatment regimens consisting of ever-increasing concentrations of tobramycin and triclosan (S1 Table). No resistant mutations were isolated from populations exposed to sudden or moderate treatment regimens. After 30 passages, two resistant populations from the gradual treatment regimen were isolated, F_1 and B_2 (Table 1), and 96 independent colonies were isolated from each mutant pool that were found to be resistant to the combination.

To determine the mutation(s) responsible for the resistance to the combination, we performed whole genome sequencing on two mutants from the F1 population and four mutants from the B2 population (Table 1). These 6 mutants all showed complete resistance to tobramycin and partial resistance to the combination compared to the ancestral strain (Fig 2). The low level of killing generated by the combination may be attributed to triclosan as all six of these mutants exhibited increased sensitivity to triclosan treatment alone versus the ancestral strain. This phenomenon, termed "collateral sensitivity," is the process by which resistance to one antimicrobial simultaneously results in increased sensitivity to unrelated antimicrobials [53].

One common feature of all six mutants was a single nucleotide polymorphism (SNP) in the *fusA1* gene encoding for elongation factor G, a protein responsible for ribosomal translocation and recycling (Fig 2) [54]. It has recently been found that mutations in the *fusA1* render *P. aeruginosa* resistant to tobramycin in clinical isolates via a unknown mechanism [55]. Another

**Table 1. Bacterial strains used in this study.**

| Strain | Characteristics | BioSample Accession # | Reference |
|---|---|---|---|
| PAO1 | *P. aeruginosa* Standard Reference Strain | NA | [30] |
| Xen41 | Bioluminescent PAO1 derivative: constitutively expresses *luxCDABE* gene | NA | PerkinElmer |
| 30_F1_5 | *fusA1*, L40Q (C**T**G→C**A**G), *wspF*, Q132* (CAG→TAG), *ptsP*, A718P (**G**CC→**C**CC), *mexT*, P170L (C**C**G→C**T**G) | SAMN15325472 SAMN15325473 | This Study |
| 30_F1_6 | *fusA1*, L40Q (C**T**G→C**A**G), *wspF*, Q132* (CAG→TAG), *ptsP*, coding (703/2280 nt), *mexT*, coding (58-137/1044 nt) | SAMN15325474 SAMN15325475 | This Study |
| 30_B2_8 | *fusA1*, T64A (**A**CC→**G**CC), *wspF*, Q132* (CAG→TAG), *ptsP*, coding (703/2280 nt), *mexT*, coding (58-137/1044 nt) | SAMN15325476 SAMN15325477 | This Study |
| 30_B2_30 | *fusA1*, T64A (**A**CC→**G**CC), *wspF*, Q132* (CAG→TAG), *ptsP*, A718P (**G**CC→**C**CC), *mexT*, P170L (C**C**G→C**T**G) | SAMN15325482 SAMN15325483 | This Study |
| 30_B2_57 | *fusA1*, T64A (**A**CC→**G**CC), *wspF*, Q132* (CAG→TAG), *ptsP*, coding (703/2280 nt), *mexT*, coding (58-137/1044 nt) | SAMN15325478 SAMN15325479 | This Study |
| 30_B2_82 | *fusA1*, T64A (**A**CC→**G**CC), *wspF*, Q132* (CAG→TAG), *ptsP*, A718P (**G**CC→**C**CC), *mexT*, P170L (C**C**G→C**T**G) | SAMN15325480 SAMN15325481 | This Study |

All additional SNPs in evolution mutants are listed in the supplemental spreadsheet. Resistant mutant naming scheme: Cycle_Location in a 96 well plate_Individual mutant number. Whole genome sequences (WGS) can be accessed at NCBI BioProject using the above accession numbers, BioProject PRJNA640554.

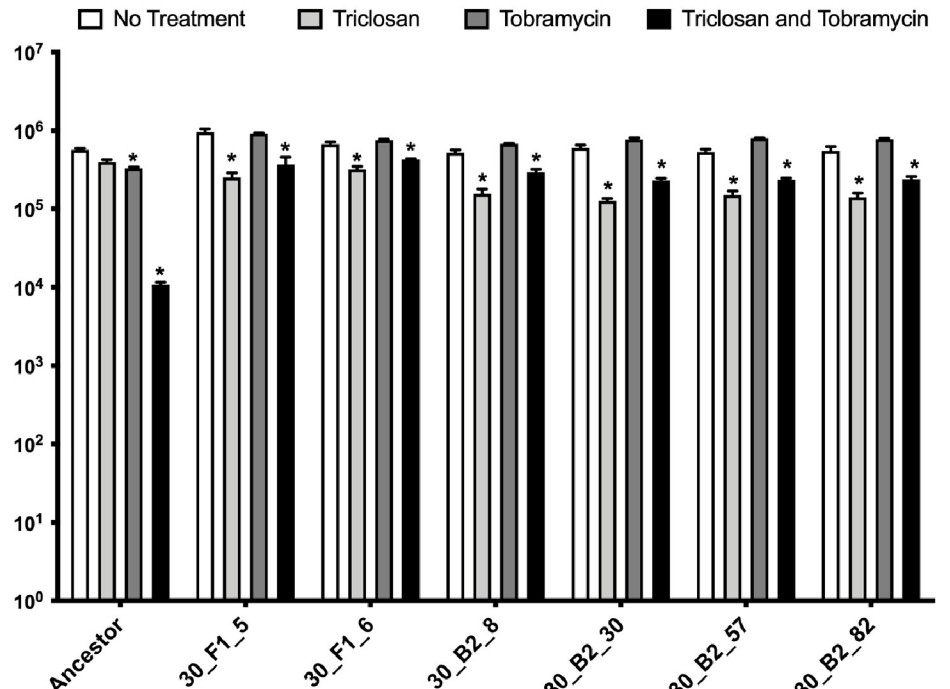

| Strain | Genotype |
|---|---|
| **30_F1_5** | *fusA1*, L40Q (CTG→CAG)<br>*wspF*, Q132* (CAG→TAG)<br>*PtsP*, A718P (GCC→CCC)<br>*mexT*, P170L (CCG→CTG) |
| **30_F1_6** | *fusA1*, L40Q (CTG→CAG)<br>*wspF*, Q132* (CAG→TAG)<br>*ptsP*, coding (703/2280 nt)<br>*mexT*, coding (58 137/1044 nt) |
| **30_B2_8** | *fusA1*, T64A (ACC→GCC)<br>*wspF*, Q132* (CAG→TAG)<br>*ptsP*, coding (703/2280 nt)<br>*mexT*, coding (58 137/1044 nt) |
| **30_B2_30** | *fusA1*, T64A (ACC→GCC)<br>*wspF*, Q132* (CAG→TAG)<br>*ptsP*, A718P (GCC→CCC)<br>*mexT*, P170L (CCG→CTG) |
| **30_B2_57** | *fusA1*, T64A (ACC→GCC)<br>*wspF*, Q132* (CAG→TAG)<br>*ptsP*, coding (703/2280 nt)<br>*mexT*, coding (58 137/1044 nt) |
| **30_B2_82** | *fusA1*, T64A (ACC→GCC)<br>*wspF*, Q132* (CAG→TAG)<br>*ptsP*, A718P (GCC→CCC)<br>*mexT*, P170L (CCG→CTG) |

**Fig 2. Mutants from experimental evolution are resistant to tobramycin and the combination.** 24-hr old biofilms were treated with triclosan (100 μM) and tobramycin (500 μM) alone and in combination. The results represent the means plus SD (n = 6). A two-way ANOVA followed by Tukey's posttest was used to determine statistical significance within each group compared to the untreated control for each strain. p<0.05.

common SNP identified in all six clones that could contribute to increased tobramycin tolerance was found in the *wspF* gene encoding for a regulator of the diguanylate cyclase WspR [56–58]. Mutations that inhibit function of the *wspF* gene result in increased intracellular cyclic di-GMP stimulating extracellular polymeric substances (EPSs) synthesis, resulting in the formation of rugose small-colony variants (RSCVs), which are significantly more tolerant to tobramycin due to reduced diffusion [56–58]. Finally, mutations in the *mexT* gene, a repressor of the RND-type efflux pump MexEF-OprN, were isolated in all six isolates, likely resulting in overexpression of this pump due to loss of repression [59]. Overall, no two isolates have the exact same set of SNPs, indicating the sequenced strains are not siblings, but each isolate is related (Fig 2). Regardless of the mechanism, experimental evolution clearly demonstrates these evolved isolates have more resistance to tobramycin but decreased resistance to the combination (Fig 2), suggesting that killing by tobramycin is essential for the combination to be effective.

## Initial tobramycin treatment is required for the combination to be effective

The corruption of translation is thought to be the first step in aminoglycoside induced adaptive resistance [24]. This led us to hypothesize that tobramycin must first corrupt translation thereby triggering adaptive resistance and the induction of RND-type efflux pumps for the combination to be synergistic [11,18,19,24]. To test if tobramycin induction of adaptive resistance was important for synergy, we first treated *P. aeruginosa* biofilms with tobramycin alone for 3-hrs and then washed the biofilms three time in Dulbecco's Phosphate Buffer Solution (DPBS) for 3-mins each followed by subsequent treatment with triclosan alone for 3-hrs. We performed a similar experiment first treating with triclosan followed by tobramycin treatment.

Sequential treatment with tobramycin followed by triclosan addition exhibited synergy, resulting in ~2-$\log_{10}$ reduction in cells within biofilms compared to untreated controls that resembled killing with the combination added for the entirety of the experiment (S3 Fig). Intriguingly, despite no longer having tobramycin in the media, the cells within the biofilms were killed when treated with triclosan. We speculate that enough tobramycin remains after washing due to association of aminoglycosides with extracellular matrix, which has been demonstrated to be resistant to removal by washing [60]. In comparison, initial treatment with triclosan followed by treatment with tobramycin did not cause synergy or result in significant biofilm killing. These data suggest that a phenotypic switch induced by aminoglycoside exposure is important for synergistic activity.

In support of these experiments, we previously showed that tobramycin induces classic adaptive resistance, rendering *P. aeruginosa* refractory to killing by tobramycin until 6-hours, with the greatest resistance observed at 2-hrs [30]. Furthermore, we found that only antibiotics that corrupt translation (aminoglycosides and tetracycline) act synergistically or are enhanced when combined with triclosan [30]. Our past and current results lead us to hypothesize that tobramycin triggers adaptive resistance, which is then disrupted by triclosan.

## Triclosan reduces RND-type efflux pump activity

Because RND-type efflux pumps are a major component of aminoglycoside induced adaptive resistance [11, 18, 19, 24], we measured their activity in biofilms after treatment with tobramycin alone or in combination with triclosan by measuring cellular accumulation of ethidium bromide. Ethidium bromide is a substrate of RND-type efflux pumps, and its accumulation within cells can be used as a proxy for their activity [61]. In this assay, the protonophore carbonyl cyanide 3-chlorophenylhydrazone (CCCP), which is known to reduce efflux pump activity, was used as a positive control [61]. CCCP treatment led to increased ethidium bromide accumulation in *P. aeruginosa* biofilms compared to the untreated control as expected (Fig 3). Triclosan also reduced efflux pump activity but to a greater extent than CCCP as indicated by increased ethidium bromide accumulation. Tobramycin treatment alone had no effect. The combination of triclosan and tobramycin was similar to triclosan alone. These data suggest that triclosan reduces the activity of RND-type efflux pumps.

## Triclosan increased intracellular accumulation of tobramycin

Triclosan inhibition of RND-type efflux pumps should increase the accumulation of tobramycin within cells. To test this prediction, we conjugated tobramycin with the fluorescent dye Texas Red (TbTR) as previously described [62], and measured its accumulation within cells growing as biofilms. Conversely, we also measured TbTR extrusion from cells into treatment-free recovery media.

For accumulation assays, biofilms were treated with TbTR with and without triclosan for 30-mins. Based on our previously published time-killing studies, this time point is before the combination causes cell death [30]. Prior to reading fluorescence, biofilms were washed to remove TbTR in the supernatant, mechanically disrupted with a wooden stick, and cells were lysed to release TbTR. Triclosan in combination with TbTR resulted in a statistically significant increase in TbTR accumulation whereas CCCP did not significantly increase TbTR accumulation (Fig 4A). This is in agreement with CCCP's reduced potency compared to triclosan at disrupting the membrane potentials of bacteria due to differences in lipophilicity [36] and the reduced activity against RND-type efflux pumps in *P. aeruginosa* biofilms that we report here (Fig 3).

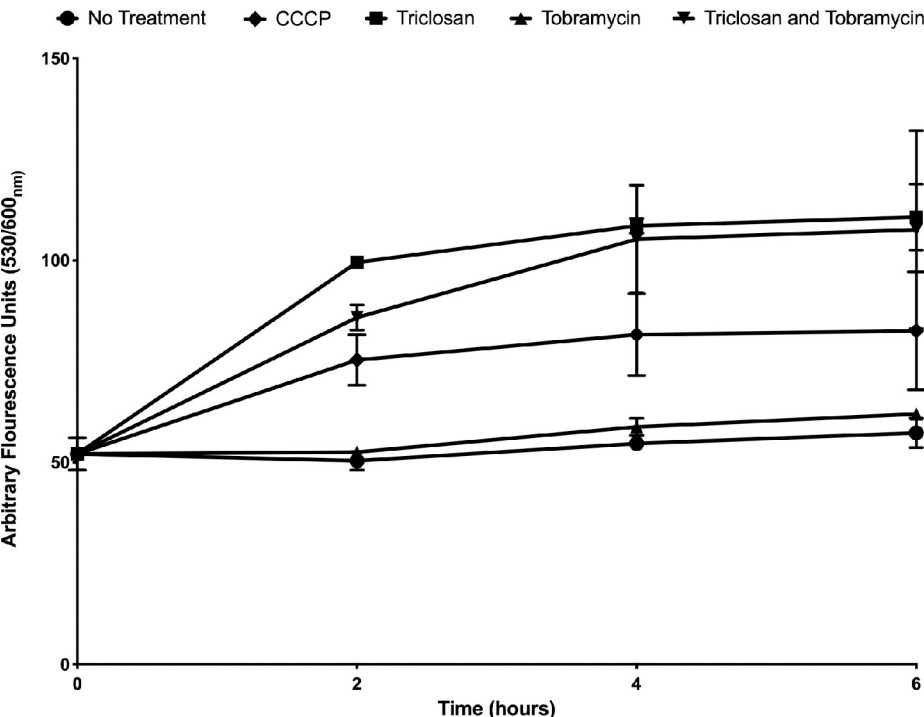

**Fig 3. Triclosan reduces the activity of RND-type efflux pumps.** 24-hr biofilms were stained with ethidium bromide to measure accumulation. Biofilms were treated with CCCP (100 μM), triclosan (100 μM), tobramycin (500 μM) alone and in combination. Fluorescence was read at 0,2,4, and 6-hrs. Results represent the average arbitrary fluorescence units ±SD (n = 6).

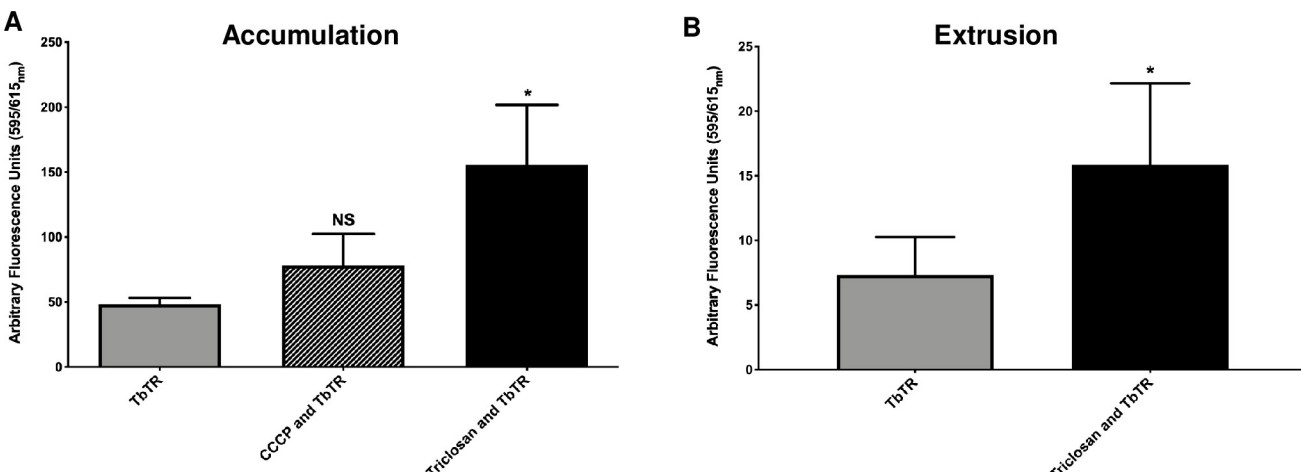

**Fig 4. Triclosan results in increased cellular accumulation and extrusion of Texas Red conjugated tobramycin (TbTR).** 24-hr old biofilms grown in glass test tubes were treated for 30-mins with triclosan (100 μM), CCCP (100 μM) and TbTR (250 μg/mL). (A) Then cells were disrupted from the biofilms and lysed with 0.2% Triton-X 100 to measure intracellular accumulation of TbTR. (B) To measure extrusion, biofilms were first treated for 30-mins and washed three-times in DPBS. Biofilms then recovered in treatment free media for 30-mins and fluorescence of the media was measured. TbTR was measured by relative fluorescence units using excitation $595_{nm}$ and emission $615_{nm}$. Results represent the average arbitrary fluorescence units plus the SD (n = 6). For panel A, a One-Way-ANOVA was performed followed by Tukey's post-hoc test was used to determine statistical significance between each group compared to TbTR alone. For panel B, an unpaired t-test was performed comparing TbTR vs triclosan with TbTR. *, p<0.05. NS, not significant.

To confirm that cotreatment with triclosan increased tobramycin accumulation within cells, biofilms were treated with TbTR with and without triclosan for 30-mins. Then biofilms were washed three-times in DPBS, recovered in treatment-free media for 30-mins and the supernatant was measured. Triclosan in combination with TbTR resulted in greater overall extrusion into treatment-free media, indicating more TbTR had accumulated in cells prior to recovery (Fig 4B). These results support the conclusion that triclosan reduces the activity of RND-type efflux pumps, leading to increased accumulation of TbTR within cells.

## Triclosan inhibits tobramycin induced Δψ

Triclosan has a hydroxyl group with a dissociable proton that can disrupt the Δψ in eukaryotic cells, mitochondria, and bacteria [34–38]. Protonophores have a dissociable proton and can pass through lipid bilayers as a conjugate base, shuttling protons across cellular membranes [63]. This activity ultimately leads to a depletion in the Δψ by reducing the charge separation across the inner membrane [39,40].

Because adaptive resistance is primarily the result of PMF driven RND-type efflux pumps [11,18,19,24], we speculated that the depletion of the Δψ could lead to their inhibition and the abolishment of adaptive resistance. In these experiments, we measured changes in the Δψ rather than the ΔpH because at neutral pH the PMF in bacteria is primarily composed of the Δψ due to homeostatic buffering [64,65], and RND-type efflux pump activity is primarily driven by the Δψ which is generated more rapidly than ΔpH [66,67]. To measure changes in the Δψ of *P. aeruginosa* growing as biofilms, cells were stained with the fluorescent Δψ indicator DiOC2(3) dye and analyzed by flow cytometry. Cells were also co-stained with the membrane impermeant TO-PRO™-3 iodide dye to exclude permeabilized cells from analysis of Δψ. The DiOC2(3) dye emits in the fluorescein isothiocyanate (FITC) channel within all cells. However, greater membrane potential drives accumulation and stacking of the dye in the cell cytoplasm, shifting emission to the phycoerythrin (PE) channel (see S4 Fig for the flow cytometry gating strategy).

Triclosan treatment for 2-hrs reduced the population of cells with a Δψ 4-fold from ~20% to ~5% (Fig 5). Tobramycin treatment resulted in an increase in the population of cells with a Δψ almost 2-fold, from ~20% to ~37%. This increase is indicative of adaptive resistance [11,18,19,24]. Triclosan in addition to tobramycin reduced the population of cells with a Δψ almost 4-fold from tobramycin treatment alone to levels comparable to untreated biofilms. These data suggest that triclosan depletes the Δψ at an essential time in aminoglycoside induced adaptive resistance when a surge in the Δψ is protective against tobramycin by increasing RND-type efflux pump activity.

## Methyl-triclosan does not disrupt the membrane potential or synergize with tobramycin

To confirm that protonophore activity by triclosan is responsible for tobramycin enhancement, we treated cells with the triclosan analog methyl-triclosan, which lacks the hydroxyl moiety necessary for gaining and losing a proton and therefore cannot act as a protonophore [35]. In addition, without this hydroxyl group, methyl-triclosan cannot inhibit fatty acid synthesis by forming a complex with the co-factors NADH/NAD+ [35]. Because our results from Fig 1 and a previous publication demonstrate that inhibition of FabI is not the mechanism of triclosan synergy [30], methyl-triclosan allows us to chemically test if disruption of the membrane potential by triclosan is necessary for synergy with tobramycin.

We first performed biofilm susceptibility testing using methyl-triclosan alone and in combination with tobramycin. Neither methyl-triclosan nor the combination were effective against

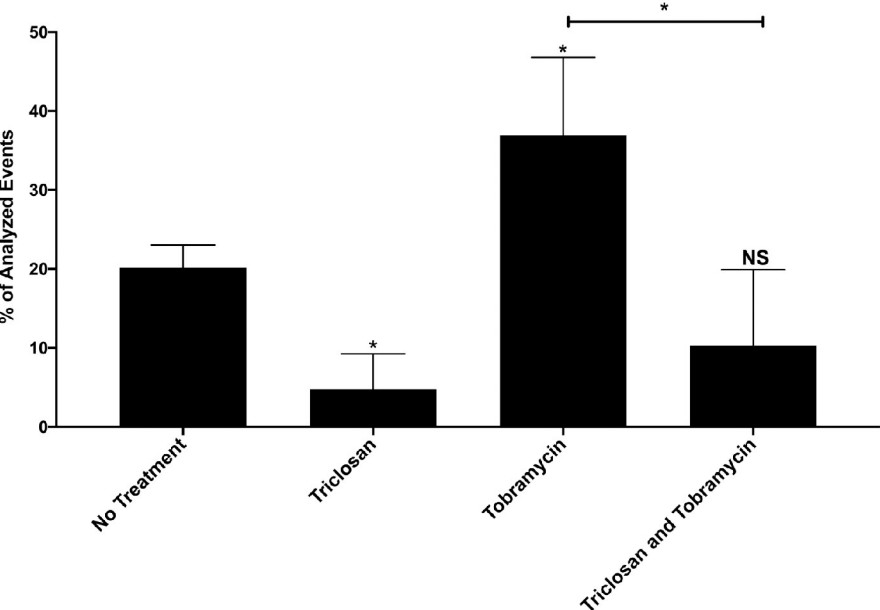

**Fig 5. Triclosan reduces the membrane potential surge induced by tobramycin at 2-hours.** 24-hr old biofilms were treated with triclosan (100 μM), or tobramycin (500 μM), alone and in combination for 2-hrs. Cells were stained with DiOC2(3) to measure the membrane potential. Dead or permeabilized cells were excluded from membrane potential analysis by parallel staining with the dead cell indicator TO-PRO™-3 iodide. The results are percent averages plus the SD (n = 4). Percent values indicate the average relative abundance of events within each gate normalized to the total number of events analyzed, excluding dead cells, artifacts, aggregates and debris. A one-way ANOVA followed by Dunnett's multiple comparison post-hoc test was used to determine statistical significance between each treatment and the untreated control, and Bonferroni's post-hoc test was performed to compare tobramycin vs tobramycin and triclosan. *, $p < 0.05$. NS, not significant.

biofilms after 6-hrs of treatment over a range of concentrations (S5 Fig). Furthermore, methyl-triclosan had no effect on the population of cells maintaining a Δψ compared to untreated biofilms and did not reduce the tobramycin induced surge in Δψ after 2-hrs of treatment (S6A Fig). In addition, methyl-triclosan did not cause permeabilization of cells alone or in combination with tobramycin (S6B Fig). Likewise, methyl-triclosan was unable to significantly reduce efflux pump activity (S7 Fig). These results support the conclusion that triclosan protonophore activity is required for the depletion of the tobramycin induced Δψ surge and inhibition of RND-type efflux pump activity.

## ATP Depletion does not render cells more sensitive to tobramycin

Another outcome of triclosan depletion of the Δψ would be decreased energy generation, which could contribute to the sensitivity of biofilms to tobramycin. To assess this potential activity, we performed antimicrobial susceptibility assays on 24-hr old biofilms grown in rich medium that were then DPBS-starved for 5 days. A similar experiment incubating *P. aeruginosa* biofilm in nutrient poor conditions demonstrated a decrease in ATP levels without a loss of viable bacteria [68]. Consistent with this previous study, we observed that the levels of ATP in the untreated biofilm were lower in the five day DPBS incubated biofilms compared with 24 hour biofilms as determined by BacTiterGlo analysis while the colony forming units (CFUs) were not significantly different (S8 Fig). However, we observed the same synergy of the combination in these conditions as treatment with triclosan or tobramycin alone failed to cause bacterial killing whereas the combination of triclosan and tobramycin caused significant killing

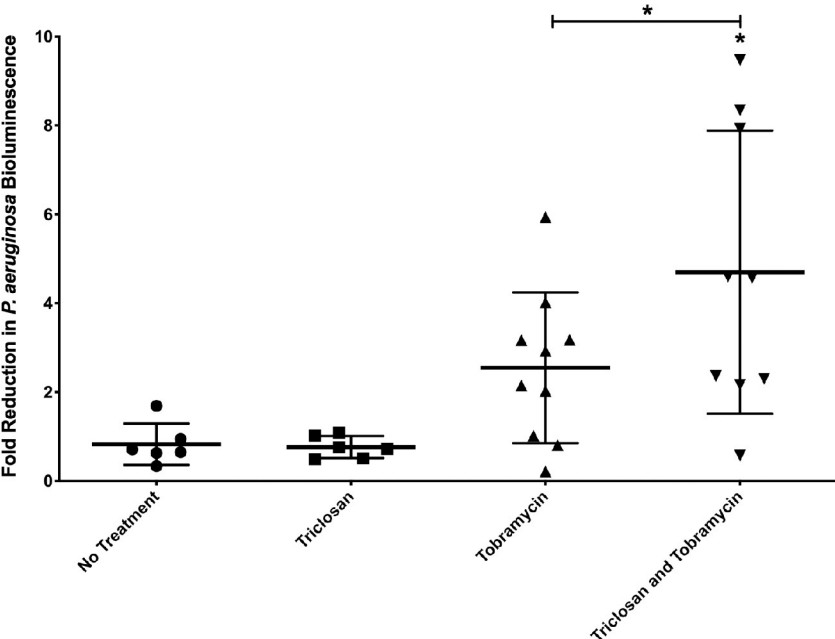

**Fig 6. Triclosan and tobramycin hydrogels are more effective than tobramycin hydrogels in a murine wound model.** 24-hr old bioluminescent biofilms formed within wounds were treated with triclosan (100 μM), or tobramycin (400 μM), alone and in combination for 4-hrs in a hydrogel. Reduction in the number of cells within biofilms was quantified using IVIS. The results are fold reduction of three separate experiments ±SD, no treatment n = 6, triclosan n = 6, tobramycin n = 10, triclosan and tobramycin n = 9. A one-way ANOVA followed by Dunnett's multiple comparison test was used to determine statistical significance between each treatment and the untreated control and, and Bonferroni's post-hoc test was performed to compare tobramycin vs tobramycin and triclosan. *, p<0.05.

(S8A Fig). These data suggest that the disruption of the Δψ by triclosan does not induce synergy by depletion of ATP.

## Triclosan and tobramycin show enhanced efficacy in an *in vivo* wound model

To determine if triclosan and tobramycin are more effective against biofilms *in vivo*, we tested their activity using a murine wound model [69,70]. In this model, a 1-day old wound on the back of SKH-1 hairless mice were infected with ~1x10$^9$ XEN 41 bioluminescent *P. aeruginosa* cells growing as biofilms to allow imaging of the infection using the In Vivo Imaging System (IVIS).

2-day old biofilms were treated for 4-hrs using an agarose hydrogel imbedded with either triclosan or tobramycin alone or in combination. Triclosan hydrogels had no effect compared to control hydrogels while tobramycin hydrogels resulted in statistically non-significant 2.5-fold-reduction in bioluminescence (Fig 6). Hydrogels embedded with triclosan and tobramycin resulted in statistically significant 4.5-fold-reduction in bioluminescence. The combination hydrogels resulted in greater than 4-fold reduction in five mice whereas tobramycin treatment only achieved this level of killing once.

## Discussion

Here we report that triclosan acts as a protonophore uncoupler against *P. aeruginosa* growing as biofilms [34–38]. To our knowledge, this is the first report of protonophore activity by triclosan against Gram-negative bacteria growing as biofilms both *in vitro* and *in vivo*.

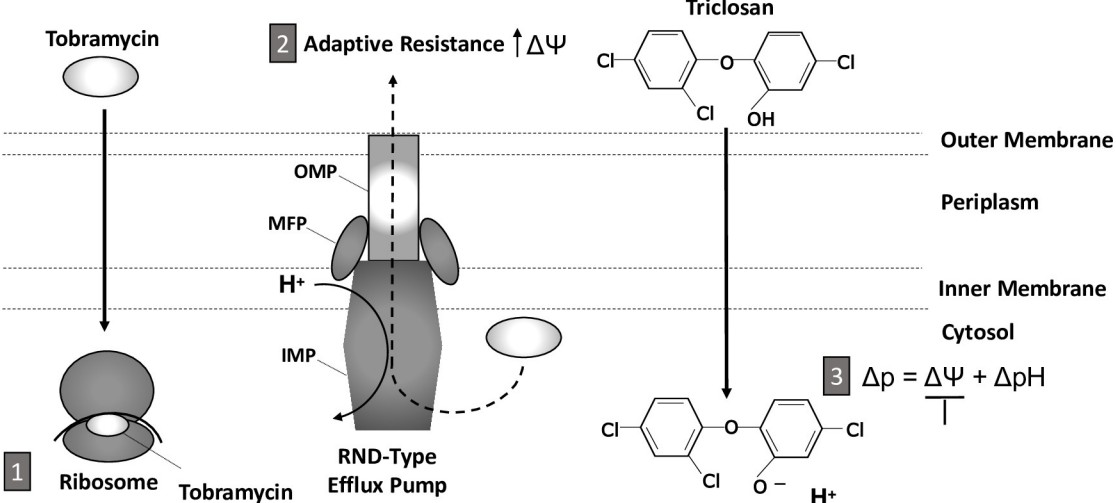

**Fig 7. Triclosan sensitizes *P. aeruginosa* to tobramycin by acting as a protonophore, inhibiting efflux pump activity, and abolishing adaptive resistance.** (1) Within 2-hrs of exposure to tobramycin, adaptive resistance occurs, (2) which is due to the induction of RND-type efflux pumps and surge in membrane potential ($\Delta\Psi$), resulting in reduced accumulation of tobramycin within the cytosol. (3) Triclosan shuttles protons across the inner membrane, collapsing the proton motive force ($\Delta p$) and depolarizing the $\Delta\Psi$. Consequently, efflux pump activity is reduced and there is enhanced accumulation of tobramycin within the cytosol. Finally, tobramycin binds to the A-site of the ribosome, corrupting protein synthesis and causing membrane permeabilization. Overall, triclosan accelerates and increase the effectiveness of tobramycin by reducing the $\Delta\Psi$ and efflux pump activity. Proton gradient, $\Delta pH$. Outer membrane protein (OMP), membrane fusion protein (MFP), inner membrane protein (IMP).

Adaptive resistance, occurring within 2-hrs of exposure, reduces accumulation of antimicrobials by inducing the activity of PMF driven RND-type efflux pumps and alterations to the OM [11,12,18,19,24]. We previously demonstrated classic aminoglycoside adaptive resistance, finding *P. aeruginosa* biofilms were refractory to tobramycin killing until ~6-hrs, with the greatest resistance observed at 2-hrs. However, the addition of triclosan with tobramycin led to rapid killing of *P. aeruginosa* biofilms [30]. Here we show that triclosan reduces the activity of PMF driven RND-type efflux pumps, resulting in greater tobramycin accumulation within cells (Figs 3 and 4AB). Triclosan accomplishes this by inhibiting a tobramycin induced surge in $\Delta\Psi$ occurring at 2-hrs (Fig 5), which leads to biofilm resistance to tobramycin as we have previously demonstrated [30]. Together, our past and present data suggest that the disruption of the $\Delta\Psi$ depletes the energy required for effective efflux pump mediated adaptive resistance (Fig 7) [30]. In support of this model, methyl-triclosan, which lacks the key hydroxyl group and thus cannot act as protonophore, was completely ineffective alone and in combination with tobramycin (S5–S7 Figs).

In addition to protein-specific corruption of translation at ribosomes, the very process of aminoglycoside uptake augments its bactericidal activity by non-specifically disrupting the OM and IM [27–29]. In the first step of uptake, positively charged aminoglycosides bind to the negative charged components of the OM, including lipopolysaccharide and phospholipids in Gram-negative bacteria. This is followed by the displacement of magnesium ions responsible for OM stabilization, promoting uptake by disrupting and increasing OM permeability [27–29]. In addition, aminoglycosides can diffuse through porins in the OM [27–29]. In EDP-I, aminoglycosides cross the IM via a mechanism that is influenced by the PMF, specifically the $\Delta\Psi$, in a concentration dependent manner [27–29]. In EDP-II, aminoglycoside cause the mis-translation of proteins by membrane associated ribosomes, resulting in the insertion of

misfolded proteins in the IM, which permeabilizes the cytoplasmic membrane accelerating its own accumulation [27–29].

Several studies have shown that aminoglycoside-mediated killing of bacteria can be enhanced by supplementing exogenous metabolites or host derived molecules, which increase cellular respiration, PMF, and the uptake of aminoglycosides through EDP-I [71–76]. However, these studies used aminoglycoside concentrations ranging from 2–25 μg/mL on cells growing planktonically [71–76]. Alternatively, the experiments described here used 250 μg/mL of tobramycin on *P. aeruginosa* biofilms, and as described we observe no synergy of tobramycin and triclosan on planktonic cells (S2 Fig) [30]. Importantly, this higher concentration of tobramycin is more clinically relevant and similar to antipseudomonal therapeutic dosing in cystic fibrosis patients [1–3,25,26].

Although our model might appear to contradict these prior studies, it has been demonstrated that at concentrations of aminoglycosides greater than 30 μg/mL, as used here, the PMF is less important for aminoglycoside uptake during EDP-I [27–29]. In fact, we observed that 250 μg/mL tobramycin can ultimately lead to biofilm killing at 24-hrs similar to the combination, suggesting that slow uptake of tobramycin during the EDP-I phase in the absence of triclosan eventually permeabilizes the IM stimulating EDP-II uptake in most of the biofilm (S9 Fig) [4,60]. However, the addition of triclosan accelerates this process by uncoupling the PMF such that maximal killing occurs at 4-hrs, as previously described (S9 Fig) [30]. Together, our results support a model where triclosan reduction of the $\Delta\Psi$ reduces the PMF required for active efflux leading to rapid accumulation of tobramycin in the cytoplasm and the stimulation of EDP-II uptake, which triggers irreversible cytosolic tobramycin accumulation and cell killing (Fig 7)[4]. In support of our model, other publications have demonstrated that bacteria can be sensitized to antibiotics by molecules that dissipate the PMF [43–46,77–80]. Specifically, Verstraeten et al., found that depleting the $\Delta\Psi$ by overexpressing Obg in *E. coli* increased intercellular accumulation of tobramycin, likely through reduced efflux [79].

RND-type efflux pumps consist of three proteins which span the inner membrane and outer membrane (Fig 7) [67]. The inner membrane protein (IMP, e.g. MexX) catalyzes the H$^+$ dependent efflux of compounds and provides antimicrobial specificity whereas the periplasmic membrane fusion protein (MFP, e.g. MexY) connects the outer membrane protein (OMP, e.g. OprM) to the IMP creating an exit channel [67]. There are 12 RND-type efflux pumps encoded in the genome of *P. aeruginosa* [4] and five are well characterized: MexAB-OprM, MexC-D-OprJ, MexEF-OprN, MexXY-OprM and MexJK-OprM [4], and two are constitutively expressed: MexXY-OprM and MexAB-OprM [81]. In particular, the MexXY-OprM pump is responsible for aminoglycoside efflux, playing an essential role in adaptive and acquired resistance [11,12,19,24,82]. The most frequent class of aminoglycoside resistant mutants in CF isolates of *P. aeruginosa* have lost function of the repressor MexZ, leading to overproduction of the RND-type efflux pump MexXY [83,84]. We previously demonstrated that the combination of tobramycin and triclosan exhibited significant killing against a clinical CF *P. aeruginosa* isolate, AMT0023_34, that is tobramycin resistant by such a mechanism [30]. Importantly, our evidence suggest that the triclosan addition can potentiate tobramycin to treat *P. aeruginosa* infections that are tobramycin resistant due to the overexpression of efflux pumps. Which of the multiple efflux pumps whose activity is negatively impacted by triclosan addition is currently under investigation.

The role of efflux pumps in antimicrobial resistance of *P. aeruginosa* biofilms has been debated. Initial studies from nearly 20 years ago found MexAB-OprM, MexCD-OprJ, MexE-F-OprN and MexXY-OprM systems were not highly expressed in *P. aeruginosa* biofilms [85]. However, these experiments showed heterogenous expression with cells closest to the substrate demonstrating the highest expression of these efflux pump systems [85], which is expected in

biofilms due to biologically distinct subpopulations and the fact that the expression of MexC-D-OprJ and MexEF-OprN are induced by their substrates [86]. Contradicting these results, subsequent studies found efflux pumps are essential for both biofilm formation and tolerance. For example, MexAB-OprM and MexCD-OprJ are required for biofilm formation in the presences of macrolides [87] and MexCD-OprJ and MexAB-OprM are biofilm-specific defense mechanisms against azithromycin and colistin, respectively [88,89]. The role of RND-type efflux pumps in biofilm antibiotic tolerance was confirmed by a transposon mutant screen, which found the PA1874-1877 operon was responsible for biofilm-specific resistance to antibiotics and the expression of PA1874 is 10-fold higher in biofilms than planktonic cells [90]. Further, Sauer and colleagues found a molecular link between efflux pump expression and the biofilm phenotype [91–93]. They conducted several studies, finding that the BrlR transcription factor responds to changes in the concentrations of c-di-GMP [94,95] and is required for the maximal expression of MexAB-OprM and MexEF-OprN efflux pumps in biofilms [91–93]. However, work by Folsom and Stewart et al. found no evidence that efflux pumps promote biofilm antibiotic tolerance in *P. aeruginosa* [96,97]. It is likely these contradictory findings highlight the complex nature of phenotypic heterogeneity in biofilms, and biofilm physiology is highly dependent on experimental conditions.

Previously and reported here, we found triclosan does not synergize with tobramycin (or other aminoglycosides) against *P. aeruginosa* growing in exponential phase (S2 Fig)[30]. Importantly, cells in exponential phase are highly susceptible to aminoglycosides and were eradicated by low μM of tobramycin, a finding in agreement with *P. aeruginosa* biofilms being ∼1000-fold more tolerant to tobramycin than their planktonic counterparts [98]. Alternatively, we found that the addition of triclosan enhanced tobramycin activity against cells in stationary phase [30], which is consistent with stationary cells being more phenotypically similar to biofilms (high cell density and low metabolic state) than cells in exponential phase [99]. We hypothesize that because planktonic cells do not encounter the antimicrobial concentration gradients experienced by biofilm growing cells [100] and are rapidly killed without the required time for adaptive phenotypic changes to occur triclosan is unable to enhance tobramycin activity [4,18]. Furthermore, additional tolerance mechanisms found in cells growing as biofilms and cells in stationary phase that are not present in exponentially growing cells may be targeted by the combination.

We show that triclosan reduces efflux pump activity through the reduction of the $\Delta\psi$, but this activity does not potentiate the cells to triclosan itself, even though the RND-type efflux pump TriABC is a triclosan resistance mechanism expressed by *P. aeruginosa* [33]. We hypothesize that although triclosan reduces efflux pump activity, this reduction is not sufficient to render *P. aeruginosa* sensitive to triclosan because this species also encodes the triclosan resistant FabI ortholog FabV [32]. In addition, if triclosan reduces efflux pump activity, one may predict it should broadly enhance multiple classes of antibiotics [101]. However, our prior study demonstrated that triclosan only enhanced aminoglycosides and to a lesser extent, tetracycline [30]. Because triclosan does not completely abolish the $\Delta\psi$, we hypothesize that triclosan only enhances antibiotic killing by antibiotics that are known to trigger adaptive resistance through disrupting translation, such as aminoglycosides and tetracyclines [8,11,18–21,24].

Owing to decades of overuse, the FDA has restricted the use of triclosan due to concerns over bioaccumulation and the potential for induction of resistance to other antibiotics in bacteria. However, triclosan maintains FDA approval for use in Colgate total toothpaste at 100-times the concentrations used in this study. In support of this decision, numerous toxicity studies and the Scientific Committee on Consumer Safety published by the European Union has found that triclosan is safe when used appropriately [47–49]. Furthermore, envisioning

triclosan as an adjuvant for tobramycin therapy for CF, we previously performed acute and long-term triclosan toxicity studies in rats and showed that direct delivery of triclosan to lungs did not elicit significant toxicity [30].

More broadly, our results suggest that protonophores could be potent antibiotic adjuvants that enhance antibiotic killing against bacterial biofilms. Protonophores are being developed as novel antibiotics to target *Mtb*. Specifically, diphenyl ethers such as SQ109 are in phase II clinical trials in humans [43–46,78]. Moreover, a HTS recently identified that the c-Jun N-terminal kinase inhibitor SU3327, renamed halicin, dissipates the ΔpH and PMF, demonstrating growth inhibitory properties both *in vitro* and *in vivo* against several pathogens including some multi-drug resistant strains [80]. Importantly, protonophore uncouplers are already routinely used for parasitic infections, including oxyclozanide, which we previously found synergizes with tobramycin much like triclosan [41]. In addition to protonophores functioning as antimicrobials, there is also renewed interest in using these compounds to target such diseases as diabetes and cancer, as well as slow aging, suggesting compounds with this activity are druggable targets when used appropriately [102–104]. Together, our results and other published studies demonstrate that the use of protonophore uncouplers to target bacterial energetics is a promising new strategy to target antimicrobial resistant infections [43].

## Methods

### Ethics statement

Vertebrate research was approved by the Michigan State University Institutional Animal Care and Use Committee application 03/18-036-00. Michigan State University is accredited by the Association for Assessment and Accreditation of Laboratory Animal Care (AAALAC).

### Bacterial strains, culture conditions, and compounds

All strains used in this study are listed in Table 1. Bacterial strains were grown in glass test tubes (18 X 150 mm) at 35˚C in cation adjusted Müeller-Hinton Broth II (MHB, Sigma-Aldrich) with agitation at 210 revolutions per minute (RPM). Antibiotics, methyl-triclosan, carbonyl cyanide 3-chlorophenylhydrazone (CCCP), and triclosan were obtained from Sigma-Aldrich. AFN-1252 was obtained from MedChemExpress. Tobramycin sulfate, gentamicin sulfate, and streptomycin sulfate were dissolved in autoclaved deionized water and filter sterilized using 0.22 μM filter membranes (Thomas Scientific). Triclosan was dissolved in 100% ethanol. Methyl-triclosan, CCCP, and AFN-1252 were dissolved in 100% dimethyl sulfoxide (DMSO).

### Biofilm susceptibility testing using BacTiter-Glo™

To measure the antimicrobial susceptibility of biofilms, the MBEC Assay (Innovotech) was used as described [30]. Briefly, an overnight culture was diluted and seeded into a MBEC plate in 10% MHB v/v diluted in DPBS and incubated for 24-hrs at 35˚C with agitation at 150 RPM. The MBEC lid was then washed to remove non-adherent cells, transferred to a 96-well treatment plate, and incubated for the indicated time at 35˚C without agitation. Following treatment, the MBEC lid was washed and transferred to a black 96-well ViewPlate (PerkinElmer) filled with 40% (v/v) BacTiter-Glo™ (Promega) diluted in DPBS to enumerate cell viability using luminescence by an EnVison Multilabel Plate Reader (PerkenElmer, Waltham, MA). The BacTiter-Glo™ Microbial Cell Viability Assay is a luminescent assay that determines the number of viable cells based on quantification of ATP concentration. A calibration curve was previously performed and it was found using a linear regression the coefficient of determination was $r^2 = 0.9884$ for luminescence versus CFUs/mL [30].

## Selection for *P. aeruginosa* resistant mutants

To select mutants resistant to tobramycin and triclosan, we modified a protocol by Lindsey and collages and serially passaged biofilms while treating them with ever increasing concentrations of triclosan and tobramycin (S1 Table) [52]. 24-hr biofilms were formed as described above and treated with triclosan and tobramycin for 24-hrs. After each treatment, biofilms were sonicated for 15-mins (Branson Ultrasonics) to disperse surviving cells from the pegs into 10% v/v MHB recovery media. Biofilms were then allowed to re-form on new pegs overnight. The next day, passaged biofilms were treated at a slightly higher concentration of triclosan and tobramycin. Following the treatment, the recovery process was repeated, and biofilms were re-formed. We split the treatment groups into three regimens: gradual, moderate and sudden (S1 Table). The gradual treatment group was initially treated with 0.4 µM of triclosan and 0.004 µM of tobramycin, which is 250x less than the MIC of tobramycin [30]. The moderate treatment group was initially treated with 8 µM of triclosan and 0.08 µM of tobramycin, which is 12.5x less than the MIC of tobramycin [30]. And then the two treatment groups were treated in subsequent cycles with ever increasing concentrations of triclosan and tobramycin at the same rate. The sudden treatment series was treated with 500 µM of triclosan and tobramycin from cycle day 1 and was eradicated. By serially passaging biofilms and gradually increasing the concentration of triclosan and tobramycin, two resistant mutant populations were established, 30_B2 and 30_F1 (Cycle_Location on 96-well plate_Individual mutant number). Each mutant pool was then streaked on LB agar plates for single colony isolation, and 96 single colony mutants were isolated from each plate, totaling 192 single colony mutants.

## Whole genome sequencing

Genomic deoxyribonucleic acid (gDNA) was isolated from each mutant using the Wizard gDNA purification kit (Promega). Illumina NextSeq was then performed by the Genomic Services Facility at Indiana University Center for Genomics and Bioinformatics. To identify single nucleotide polymorphisms (SNPs), sequencing results were first verified for quality using FASTQC and aligned to the PAO1 reference genome [105], which can be downloaded from the *Pseudomonas* Genome Database (http://www.pseudomonas.com) using the *Breseq* pipeline, which can be downloaded from (http://barricklab.org) [106]. All identified SNPs are listed in the supplemental spreadsheet. Whole genome sequences (WGS) can be accessed at NCBI BioProject PRJNA640554 and the accession numbers for each genome sequence is listed in Table 1.

## Ethidium bromide efflux pump assay

Intercellular accumulation of ethidium bromide, which is a substrate for RND-type efflux pumps, was measured fluorometrically as previously described [61,107]. As described above, 24-hr old biofilms were formed in 96-well black ViewPlate. 10 µg/mL of ethidium bromide was added to each well along with various treatments. Fluorescence was recorded every 2-hrs for 6-hrs using a SpectraMax M5 microplate spectrophotometer system ($\lambda_{excite}/\lambda_{emit}$ 530/600 nm).

## BacLight™ membrane potential assay and cell permeabilization assay

To measure changes in membrane potential and cellular permeabilization the BacLight™ Membrane Potential Assay was used in combination with the cell impermeable live/dead TO-Pro™-3 iodide stain, as previously described [41]. Briefly, 24-hr old biofilms were formed in glass test tubes (18 x 150 mm) in 1 mL of 10% (v/v) MHB at 35°C and agitated at 150 RPM. Cells were washed in DPBS to remove non-adherent cells and treated with triclosan and tobramycin for 2-hrs. Following treatment, cells were washed in DPBS (without magnesium and calcium),

and the biofilm was disrupted from the air-liquid interface. The cells were stained in 1 mL of DPBS for 5-mins with DiOC2(3). This dye emits in the fluorescein isothiocyanate (FITC) channel within all cells. However, greater membrane potential drives accumulation and stacking of the dye in the cell cytoplasm, shifting emission to the phycoerythrin (PE) channel. TO-PRO™-3 iodide, which emits in the allophycocyanin (APC) channel, was used to identify permeabilized cells, which fluoresces in cells that have compromised membranes by intercalating DNA. Single cell flow cytometry was performed on an LSR II (BD Biosciences), with excitation from 488 mm and 640 mm lasers, and analyzed in FITC/PE and APC channels, respectively. Representative gating strategy can be found in S7 Fig.

### Tobramycin accumulation and extrusion assay

To measure the accumulation of tobramycin within cells in biofilms, tobramycin was conjugated to Texas Red (Sigma-Aldrich) using an amine conjugation reaction, as previously described [62]. Briefly, one milligram of Texas Red sulfonyl chloride (Thermo Fisher Scientific) was resuspended in 50 µL of anhydrous N,N-dimethylformamide (Sigma-Aldrich) on ice. The solution was added slowly to 2.3 mL of 100 mM K2CO3 at pH 8.5, with or without 10 mg/mL tobramycin (Sigma-Aldrich), on ice. Conjugated tobramycin was used at a concentration of 250 µg/mL (~500 µM) alone and in combination with 100 µM of triclosan against 24-hr old biofilms formed in glass test tubes at the air-liquid interface, as described above. Following treatments, biofilms were washed in DPBS and then disrupted with autoclaved wooden sticks into 1 mL of 0.2% Triton X-100 to lyse cells (Sigma-Aldrich). Lysed cells were then transferred to spectrophotometer cuvettes (Thermo Fisher Scientific) and read using a SpectraMax M5 microplate spectrophotometer system ($\lambda_{excite}/\lambda_{emit}$ 595/615 nm). Extrusion assays were performed in a similar fashion, except after 30-mins of treatment, biofilms were washed three times in DPBS and then allowed to recover in treatment-free 1% MHB media for 30-mins. After recovery, the media was read as described above.

### DPBS-starved *P. aeruginosa* biofilms

24-hr old biofilms were formed on MBEC™ plates as described above. After 24-hrs, the lid was washed for 5-mins to remove non-adherent cells, transferred to a 96-well plate containing DPBS, and incubated at 35°C with agitation at 150 RPM for 5-days. CFUs were determined by removing the MBEC™ plate pegs with needle nose pliers into 500 µL of MHII followed by vortexing for 30 seconds and dilution plating.

### Agarose hydrogels

Agarose hydrogels were made as previously described [70] by dissolving 1 gm of agarose (Sigma-Aldrich) into 200 mL of Tris-acetate-EDTA (TAE) buffer and heated to form a homogenous solution using a microwave. The 0.5% agarose solution was then allowed to cool and various treatments were added. The solution was then poured into 100 x 15 mm petri dishes (Thermo Fisher Scientific) and stored at 4°C overnight. Prior to treatment, a 4 mm biopsy punch (VWR) was used to create hydrogel wafers.

### Murine wound infection model

Wound surgery was performed on 8–9 week-old male and female SKH-1 mice (Charles River) and infected with a bioluminescent derivative of PAO1, Xen41 (PerkinElmer), which constitutively expresses *luxCDABE* gene, as previously described with the following modifications [69, 70]. Briefly, 24-hr old biofilms were formed on sterilized polycarbonate membrane filters with

a 0.2 μM pore size (Millipore Sigma) by diluting an overnight culture to an $OD_{600}$ of 0.001 and pipetting 100 μl on 4 membranes on a tryptic soy agar (TSA) plate. 24-hr old biofilms were scrapped using L-shaped spreaders (Sigma-Aldrich) from each membrane and re-suspended in 500 μl of DPBS. 20 μl of the biofilm-suspension was inoculated into the 24-hr old wounds previously formed on the dorsal side of the mouse midway between the head and the base of the tail. 24-hrs later the biofilm within the wound was imaged using *in vivo* imaging system (IVIS) (Perkin Elmer). Then the biofilm was treated by placing a 4 mm 0.5% agarose hydrogel wafer either containing various treatments or only being made up of agarose alone on top of the wound. To quantify antimicrobial activity, the wound was then imaged 4-hrs later using IVIS and total flux (p/s) in radiance was used to quantify bacterial susceptibility. The loss of radiance indicates fewer cells are present within the wound. Vertebrate research was approved by the Michigan State University Institutional Animal Care and Use Committee application 03/18-036-00.

## Statistics

Significance was determined as described in accompanying figure legends by either a one-way ANOVA or two-way ANOVA followed by Tukey's, Dunnett's or Bonferroni's multiple comparisons posttest, respectively. Unpaired t-test were also used as described in accompanying figure legends. The "n" indicates the number of biological replicates. For murine wound infection experiments, sample sizes were determined based on preliminary studies using groups that were sufficient to achieve a desired power of 0.85 and an alpha value of 0.05. All analyses were performed using GraphPad Prism version 8.4.1 (San Diego, CA) and a *p*-value of $< 0.05$ was considered statistically significant.

## Supporting information

**S1 Fig. AFN-1252 alone or in combination with tobramycin is not effective against mature biofilms.** 24-hr old biofilms grown on MBEC plates were treated for 6-hrs with AFN-1252 (400 μM), tobramycin (400 μM), alone and in combination in two-fold dilutions, and the number of viable cells within the biofilms were quantified by BacTiter-Glo[TM]. The assay was performed twice in in duplicate. The results represent means ± the standard deviation (SD). A one-way analysis of variance (ANOVA) followed by Bonferroni's multiple comparison post-hoc test was used to determine statistical significance between tobramycin versus the combination (NS, not significant).
(PPTX)

**S2 Fig. Triclosan does not enhance tobramycin killing of planktonic cells.** Planktonic cells were treated with the indicated concentrations of tobramycin with or without 100 μM triclosan and grown for 16 hours before measuring the $OD_{600}$. Error bars indicate the standard deviation.
(PPTX)

**S3 Fig. Initial tobramycin treatment is required for the combination to be effective.** 24-hr old biofilms grown on MBEC plates were treated sequentially. First, biofilms were treated for 3-hours with tobramycin (500 μM) and then washed three times in DPBS for 3-mins each, before being treated with triclosan (100 μM) for 3-hours or vice versa. As a control, biofilms were also treated for 6-hrs with triclosan and tobramycin. The number of viable cells within the biofilms were quantified by BacTiter-Glo[TM]. The assay was performed twice in in duplicate. The results represent means plus the SD. A one-way ANOVA followed by Tukey's multiple comparison post-hoc test was used to determine statistical significance between each

treatment and the untreated control. *, p<0.05. NS, not significant.
(PPTX)

**S4 Fig. Representative flow data and gating strategy.** Untreated 24-hr old biofilms were stained with TO-PRO™-3 to determine the number of cells that were permeabilized. Cells were also stained with DiOC2(3) to determine the number of cells maintaining a membrane potential. **A.** Scatter plot of side scatter area (SSC-A) versus forward scatter area (FSC-A). Gate P1 excludes debris and artifacts. **B.** Histogram plot of count versus APC from P1 gate shown. Gate P2 indicates cells that are TO-PRO™-3 negative because they have an intact outer membrane (OM), preventing the diffusion of TO-PRO™-3 into the cell. Gate P3 indicates cells that stain positive for TO-PRO™-3, indicating permeabilization, thus allowing TO-PRO™-3 to diffuse into the cytosol and bind DNA emitting in the APC channel. **C.** Cells with intact OMs from population P2 were plotted as a scatter plot of PE-A versus FITC-A. Gate P4 indicates cells that stain positive with DiOC2(3), which emits in the FITC channel in all cells. Gate P5 indicates cells with higher membrane potentials, which drives the accumulation of DiOC2(3) within the cytosol, subsequently resulting in self-association and a shift in fluoresces to the PE channel.
(PPTX)

**S5 Fig. Methyl-triclosan alone or in combination with tobramycin is not effective against mature biofilms.** 24-hr old biofilms grown on MBEC plates were treated for 6-hrs with methyl-triclosan (100 μM), tobramycin (400 μM), alone and in combination in two-fold dilutions, and the number of viable cells within the biofilms were quantified by BacTiter-Glo™. The assay was performed twice in in duplicate. The results represent means ±SD. A one-way ANOVA followed by Bonferroni's multiple comparison post-hoc test was used to determine statistical significance compared to tobramycin alone and the combination. NS, not significant.
(PPTX)

**S6 Fig. Methyl-triclosan alone and in combination with tobramycin does not reduced the surge in membrane potential induced by tobramycin at 2-hours.** 24-hr old biofilms were treated with methyl-triclosan (100 μM), or tobramycin (500 μM), alone and in combination for 2-hrs. Cells were stained with DiOC2(3) and TO-PRO™-3 iodide to determine the number of cells that maintained a membrane potential (A) or were permeabilized (B), respectively. Dead or permeabilized cells were excluded from membrane potential analysis and are shown in panel B. The experiment was performed two separate times in duplicate. The results are percent averages plus the SD. Percent values indicate the average relative abundance of events within each gate normalized to the total number of events analyzed, excluding artifacts, aggregates and debris. A one-way ANOVA followed by Dunnett's multiple comparison post-hoc test was used to determine statistical significance between each group and the untreated control and Bonferroni's post-hoc test was performed to compare tobramycin vs tobramycin and methyl-triclosan. *, p<0.05. NS, not significant.
(PPTX)

**S7 Fig. Methyl-triclosan does not reduce RND-type efflux pump activity.** Ethidium bromide is a substrate of RND-type efflux pumps. 24-hr biofilms were stained with ethidium bromide to measure accumulation. Biofilms were treated with methyl-triclosan (100 μM) and tobramycin (500 μM) alone and in combination. Fluorescence was read at 0,2,4, and 6-hrs. The assay was performed three times in triplicate. Results represent the average arbitrary fluorescence units ±SD.
(PPTX)

**S8 Fig. DPBS starved biofilms are not more sensitive to tobramycin alone.** 24-hour old biofilms were starved of nutrients by replacing media with DPBS for 5-days. Starved biofilms were treated with triclosan (100 μM), or tobramycin (500 μM), alone and in combination for 6-hrs. The number of viable cells within the biofilms was quantified using the BacTiter-Glo™ assay. The assay was performed once using three biological replicates. The results represent the means plus the SEM. A one-way ANOVA followed by Sidak's multiple comparison post hoc test was used to determine statistical significance between the combination treatment and the untreated control and between triclosan alone or tobramycin alone and the combination treatment as indicated by the bars. $^*$, $P < 0.05$. Colony forming units (CFUs)/mL were calculated for 24-hr biofilms and biofilms that had been starved in DPBS for 5-days. A t-test was performed to determine the statistical difference between mature biofilms and starved biofilms, NS, not significant.
(PPTX)

**S9 Fig. Tobramycin is as effective as the combination after 24-hrs of treatment.** 24-hr old biofilms were treated for 24-hrs with triclosan (100 μM), tobramycin (500 μM), alone and in combination, and the number of viable cells within the biofilms were quantified by BacTiter-Glo$^{TM}$. The assay was performed at least three times in triplicate. The results represent means plus the SEM. A two-way ANOVA followed by Tukey's multiple comparison post-hoc test was used to determine statistical significance between each group. $^*$, $P < 0.05$. NS, not significant. 4-hr treatments were previously published and are shown for comparison [28].
(PPTX)

**S1 Table. 27 *P. aeruginosa* biofilms serially passaged in parallel were exposed to sudden, moderate, and gradual treatment regimens consisting of ever-increasing concentrations of tobramycin and triclosan (mM) spread out over longer and longer periods of time, as shown in the table above.** Gradual and moderate treatment groups were started at different concentrations of tobramycin and triclosan. However, they were then treated in subsequent cycles with ever increasing concentrations of triclosan and tobramycin at the same rate. For example, the moderate treatment group reached 1 μM of tobramycin and 100 μM of triclosan in 8-cycles, whereas the gradual treatment group reached the same concentration in 16-cycles from their initial treatments. The sudden treatment series was treated with 500 μM of triclosan and tobramycin from cycle day 1 and was eradicated, thus, is not shown in the table. Highlighted in yellow, the minimum inhibitory concentration (MIC) of tobramycin against planktonic cells was found to be 1μM and is shown.
(PPTX)

## Acknowledgments

We thank both Sandra O'Reilly and Louis King of the Michigan State Research Technology Support Facility for their assistance in the wound model and flow cytometry experiments, respectively. We also thank the Genomic Services Facility at Indiana University Center for Genomics and Bioinformatics. And we thank Lucas Demey, Mitchell Zachos and Emily Horning for their technical assistance.

## Author Contributions

**Conceptualization:** Michael M. Maiden, Christopher M. Waters.

**Data curation:** Michael M. Maiden, Christopher M. Waters.

**Formal analysis:** Michael M. Maiden, Christopher M. Waters.

**Funding acquisition:** Michael M. Maiden, Christopher M. Waters.

**Investigation:** Michael M. Maiden, Christopher M. Waters.

**Methodology:** Michael M. Maiden, Christopher M. Waters.

**Project administration:** Christopher M. Waters.

**Supervision:** Christopher M. Waters.

**Writing – original draft:** Michael M. Maiden, Christopher M. Waters.

**Writing – review & editing:** Michael M. Maiden, Christopher M. Waters.

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
