## [Decision Letter · Decision Letter 0]

19 May 2020

Dear Dr. Waters,

Thank you very much for submitting your manuscript "Triclosan depletes the membrane potential in Pseudomonas aeruginosa biofilms inhibiting aminoglycoside induced adaptive resistance" for consideration at PLOS Pathogens. As with all papers reviewed by the journal, your manuscript was reviewed by members of the editorial board and by several independent reviewers. In light of the reviews (below this email), we would like to invite the resubmission of a significantly-revised version that takes into account the reviewers' comments.

We cannot make any decision about publication until we have seen the revised manuscript and your response to the reviewers' comments. Your revised manuscript is also likely to be sent to reviewers for further evaluation.

Sincerely,

Matthew Parsek, PhD

Associate Editor

PLOS Pathogens

Denise Monack

Section Editor

PLOS Pathogens

Kasturi Haldar

Editor-in-Chief

PLOS Pathogens

orcid.org/0000-0001-5065-158X

Michael Malim

Editor-in-Chief

PLOS Pathogens

orcid.org/0000-0002-7699-2064

Reviewer's Responses to Questions

**Part I - Summary**

Reviewer #1: This manuscript describes a mechanism by which triclosan can sensitize Pseudomonas aeruginosa biofilms to tobramycin, which is an interesting potential therapeutic avenue. While others have shown that ionophores increase antibiotic efficacy, these results clearly show that triclosan has this effect, leading to a reduction in the membrane potential, an increase in the cellular accumulation of RND-efflux-pump clients (like tobramycin), and an increase in the tobramycin-mediated killing of P. aeruginosa in a murine wound model. The major weaknesses identified are minor elements of work and could be addressed via editing of the manuscript. To improve the significance of this work, a clearer discussion of why the authors think this mechanism is biofilm specific would be useful.

Reviewer #2: The manuscript title very accurately describes this interesting study where efficacy of tobramycin upon P. aeruginosa biofilms is enhanced by the addition of triclosan. The work is very interesting and exciting. Most importantly, the authors present a series of well-reasoned experiments to discern the mechanism of triclosan action to allow for the observed effect.

The introduction is extremely clearly worded and written. The Discussion is also well thought out, plausible, and draws from many prior studies in a very thorough manner.

Reviewer #3: In this manuscript, Maiden and Waters investigate the mechanism by which triclosan potentiates tobramycin against Pseudomonas aeruginosa biofilms. The primary finding is that triclosan affects the function of RND-type efflux pumps that expel antibiotics from the bacterial cell. The authors conclude that efflux pump inhibition results from the uncoupling of the delta-psi across the bacterial cytoplasmic membrane. A strength of the work lies in potential application of the findings to treatment of infections and that this strategy is effective against biofilms. Work from an animal model is presented. This manuscript is a follow up on previous results from the Waters lab reporting the discovery that triclosan is antibiotic adjuvant that increases the activity of aminoglycosides against biofilms (Maiden et al, 2018, Antimicrob Agents Chemother 62:e00146-18).

**Part II – Major Issues: Key Experiments Required for Acceptance**

Reviewer #1: In the initial finding by these authors, they show that triclosan is synergistic with tobramycin but only under biofilm conditions. There was no effect on planktonic cells. I would think that the effect of triclosan on RND efflux pumps is unlikely to be biofilm specific. Addition of why/how the triclosan effect is biofilm specific (in the model and the discussion) would improve the manuscript.

The interpretation of the whole-genome sequencing data is problematic. All 6 isolates that were sequenced from the two resistant populations fall into two classes. One class (represented by F1_5, B2_30, and B2_82) contains substitutions leading to MexT P170L, FusA1 L40Q, PtsP A718P, and WspF Q132*. The other class (represented by F1_5, B2_8, and B2_57) contains a truncated MexT (80 bp deletion), FusA1 T64A, a truncated PtsP (1 bp deletion), WspF Q132*, PilY1 Q506*, and PmrB M292T. There are likely other genomic changes among the isolates within each classes that are the same. Therefore, although these isolates are not siblings, because the genetic changes are exactly the same within a class, they are definitely related. It suggests that the seeding population contained 2 isolates with these mutations at very low levels. The authors suggest that it is the FusA1 substitution (among all the changes) that is selected for in their passage experiment (lines 163-175, 382-390). However, it seems more likely that it is the WspF substitution that is selected for, since WspF mutations lead to cells that produce more robust biofilms (which, as mentioned by the authors, can increase the tolerance of biofilms against tobramycin) and the serial passage method used will likely select for hyperbiofilm formers. Furthermore, the FusA1 substitution is not the only change that would lead to increased tobramycin tolerance. While this is mentioned in the Discussion (lines 391-405), it is important to acknowledge that there is no way to know which change(s) were selected for. The strong focus on FusA1 (in the Results and the Discussion) is not warranted. Separately, if isolates of these two classes are at a low level in the seed population, it may be the reason why no resistant populations were found in the moderate regime.

In the ATP depletion experiment, it is not clear that the authors actually reduced the amount of ATP in the cells by starving the biofilms for 5 days. The stringent response should be activated by the shift to nutrient starvation, which would prevent a large drop in ATP levels. While there is less BacTiter-Glo signal (e.g. ATP concentration) in biofilms after starvation for 5 days (comparing the no treatment samples in Figs 2 and S6), this could be due to some level of dispersal of the biofilm so that there are fewer cells in the biofilm after 5 days of starvation (with no change in the ATP concentration per cell). To determine if ATP levels are different, the BacTiter-Glo signal should be normalized to the CFU for a starved and un-starved biofilm. Furthermore, this experiment should use CFUs (or some other measure of viability that does not rely on ATP concentrations) as a measure of total biofilm amount, so as to not confuse the variables in the experiment.

It is not clear why the authors chose Bonferroni’s and Sidak’s for their post-hoc statistical analyses. Bonferroni’s (and Sidak’s) is prone to Type II errors, which may explain why the authors see no statistical significance in some cases where samples look very different. For almost all the data, the samples are independent. Furthermore, means are being compared, so a Tukey HSD, which would reduce the Type II errors, could be used. In addition, instead of using asterisks to show statistical significance, letters to denote statistical groupings would be more informative. For instance, in Fig 2, it would allow for the within treatment comparison across strains.

Reviewer #2: There are no major issues with this manuscript.

Reviewer #3: Many parts of this paper have merit. However, I have some concerns with the manuscript and some constructive criticism for the authors.

Major points:

1. The first two points are related to integration of the findings with the literature. This manuscript is centered on aminoglycoside and triclosan synergy against biofilms, and the authors propose that bacterial energetics can be targeted to increase the susceptibility of P. aeruginosa biofilms to aminoglycosides (first stated in the abstract, Lines 35-37). So, do these bacterial energetics underlie the increased resistance of biofilms to antibiotics, or does the mechanism of synergy here apply to P. aeruginosa in planktonic cells in exponential or stationary phase too? The information and distinction are important. If this phenomenon does not occur for planktonic cells, then the authors’ experiments might inform our basic understanding of biofilm microbiology. If these combinations are also effective against planktonic cells, then I would suggest that some of the writing surrounding anti-biofilm activity should be constructed more carefully.

2. There is a lack of background information providing context to understand adaptive resistance in biofilms. There are still few reports describing how adaptive stress responses contribute to biofilm antimicrobial resistance phenotypes, and yet this information seems to be important for understanding the mechanism of antimicrobial synergy here. This information may also be important for distinguishing between planktonic and biofilm susceptibility phenotypes. Could the authors please add this information to help provide the reader with the salient information?

3. The authors need to add more explanation and information to assist the reader with rationale and interpretation of the results. I would propose that it may be difficult for some readers, including perhaps some senior trainees, to understand how the results inform mode-of-action. This is a point pertaining to communication, not the science. For example (and there are other places too): A) Lines 155-162. Perhaps the authors could elaborate how different findings from these experiments can inform their interpretations, setting the reader up to follow along more easily? B) Line 241. Could the authors please describe how a “dissociable proton” from triclosan disrupts the delta-Psi of cell membranes? C) Line 249. Could the authors please provide a description of how the DiOC2(3) dye functions as an indicator of delta-psi?

4. A concern is drawing conclusions from results is the use of BacTiter-Glo. Caution is prudent. While it is established that viable cell counts correlate nicely with [ATP] for healthy cells, there are many potential reasons why this relationship might breakdown after a toxic antimicrobial exposure. It seems wise to repeat some of the key findings using viable cell counts to be sure of the results with BacTiter-Glo, especially since the mode of action of triclosan is thought to be via action on the delta-psi, which energizes ATP synthesis. In other words, there could be a decrease in intracellular [ATP] that does not correlate with an increase in log-killing. Also, since the authors are measuring [ATP], statements in the results about log-killing strike me as a leap in logic. As an alternative, one possible solution would be to make these statements in terms of observing changes in [ATP], rather than log-reductions in cell viability.

5. It seems that some thought has been put into use of statistics. However, why do the authors plot standard error of the mean (SEM) as opposed to standard deviation (SD)? SEM values will always be smaller than SD values because of how SEM is calculated. However, SD is the statistical measure used to show variation among values within experiments. The use of SEM is unusual, especially for susceptibility testing. There is some long-standing humour among statisticians about how biologists use SEM (http://pmean.com/05/StandardError.html). I am concerned that SEM might be misleading and may not be the right choice here.

6. Could the authors also expand on the background information that RND-type efflux pumps are responsible for the multidrug resistance and tolerance of P. aeruginosa biofilms as opposed to planktonic cells? Evidence dating back nearly 20 y suggests that these pumps may not be the main culprits behind the reduced tolerance of P. aeruginosa biofilms to antibiotics (ex. https://www.ncbi.nlm.nih.gov/pubmed/11353623). Additionally, a way of providing more supporting evidence for the mode of action proposed here would be to test strains with mutations in the RND-efflux pumps. Even if there is a hypersensitivity phenotype, antimicrobial synergy should be lost.

**Part III – Minor Issues: Editorial and Data Presentation Modifications**

Reviewer #1: The section on methyl-triclosan seems superfluous. Since the addition of the methyl group affects both activities of triclosan (dropping the membrane potential and inhibiting fatty acid synthesis), I do not see what this data adds to the manuscript. I also disagree that it supports the model (line 315), since it does not help differentiate between the different effects that triclosan is known to have.

There was no synergy when cells were treated with triclosan first (Fig 3), and the extrusion experiment shows approximately a 10-fold drop in both samples (with and without triclosan) during recovery (Fig 4). These data suggest that the triclosan-induced effect is quickly fixed by the cells as soon as the drug is removed from the extracellular environment. Why this would be the case is difficult to explain using the model presented. Why do the authors think that cells recover so quickly after triclosan removal?

It is not clear from the methods how the air-liquid interface biofilms are washed. Removing/exchanging the liquid underneath the biofilm would disturb the biofilm, no? Also in experiments that are not using the MBEC peg lids, how do the authors avoid the surface-liquid biofilm? In general, how is the air-liquid biofilm isolated?

Inclusion of the flow cytometry graphs with the gates in the Supplemental would be useful for interpretation of the graphical data.

Minor points of confusion

- Line 192: The use of “alternatively” is confusing, since it suggests that the sentence is an alternative explanation for why cells die after tobramycin has been removed. Perhaps “in comparison” could be used instead.

- Line 252 - 256: “maintaining a membrane potential x-fold” is difficult to parse. While it is easy to see the fold change in membrane potential between the samples in the figure, the sentences are difficult to understand because of the use of “maintaining.”

Reviewer #2: For Figure 1 and the normalization values, it is not clear what constitutes an “analyzed event”. This is the percentage of green fluorescent cells that also fluoresce red from FACS data? In the methods section, there is a lot of detail, but it is difficult to apply this to interpretation of the presented results.

Line 152: “After 30 passages, two resistant populations from the gradual treatment regimen were isolated (Table 1)…” The two populations are F1 and B2? As one compares this paragraph with Table 1, it is difficult to reconcile them as they have no common notation between them.

What is the significance of the yellow highlighting of Cycle 16 in Table S1? This appears to be a set of target concentrations that were tested, however, it is not the point at which either the gradual or moderate experiments stopped. Also, it is not clear how “gradual” and “moderate” are the best descriptors. Both terms could easily be applied to both sets if one considers either the initial concentration or the rate of change, yet moderate seems to describe initial starting concentrations while gradual describes a rate of change. Certainly “sudden” describes both accurately.

Line 267. This section is great. An excellent control for the prior triclosan experiments.

Line 279. It is not clear that the authors can specifically claim association with “ATP depletion” here. Certainly, an ability to generate ATP would be associated with these nutrient starved conditions, but it would be more reasonable to assign these 5-day DPBS experiments as “metabolically inactive” or some more general description, which would also include an ATP depleted state.

Line 312. “occurring at 2-hrs” This descriptor is correct because it is when the authors performed their experiment. However the statement is misleading in that it suggests the authors have delineated a timecourse for tricolosan action—which was not done here.

Reviewer #3: Line 86. Could the authors qualify the statement that the combination of triclosan and tobramycin is “100-times more effective.” Do you mean that there is a 100-fold decrease in MBC, or that killing at particular concentrations is 100-times greater?

Fig. 2. Could the authors put genotypes in the figure labels instead of strain names? This would be much more informative to the reader.

The authors could consider depositing genome sequencing information in Genbank and providing Bioproject or Biosample IDs in the manuscript.

PLOS authors have the option to publish the peer review history of their article (what does this mean?). If published, this will include your full peer review and any attached files.

Reviewer #1: No

Reviewer #2: No

Reviewer #3: No
---

## [Decision Letter · Decision Letter 1]

15 Sep 2020

Dear Dr. Waters,

We are pleased to inform you that your manuscript 'Triclosan depletes the membrane potential in Pseudomonas aeruginosa biofilms inhibiting aminoglycoside induced adaptive resistance' has been provisionally accepted for publication in PLOS Pathogens.

Best regards,

Matthew Parsek, PhD

Associate Editor

PLOS Pathogens

Denise Monack

Section Editor

PLOS Pathogens

Kasturi Haldar

Editor-in-Chief

PLOS Pathogens

orcid.org/0000-0001-5065-158X

Michael Malim

Editor-in-Chief

PLOS Pathogens

orcid.org/0000-0002-7699-2064

Reviewer Comments (if any, and for reference):

Reviewer's Responses to Questions

**Part I - Summary**

Reviewer #1: This revised manuscript describing the mechanism behind the synergy of triclosan and tobramycin in P. aeruginosa biofilm killing is much improved. There are no major remaining concerns. The previously noted major weaknesses (on minor elements) have been generally addressed (see minor comments are below).

Reviewer #2: This revised manuscript is much improved. It is clear the authors have done a very thorough and considerate job of addressing all prior reviewer comments. The description of changes leaves no doubt and the explanations are very clear. This remains excellent work and now the written manuscript reflects this throughout.

Reviewer #3: I think that the authors have done a terrific job in addressing the reviewer comments. I note that the authors have put considerable thought into their use of statistical tests and have made necessary corrections. The added extra data is appreciated too. I really enjoyed reading this revised manuscript – it is well written and will be well received by our community.

**Part II – Major Issues: Key Experiments Required for Acceptance**

Reviewer #1: None

Reviewer #2: No new concerns.

Reviewer #3: None.

**Part III – Minor Issues: Editorial and Data Presentation Modifications**

Reviewer #1: - Lines 448-454: In response to why the effect is biofilm specific, the authors state that it may be due to rapid killing of planktonic cells. However, this does not address why there is no synergy between triclosan and tobramycin at sub-MIC concentrations of tobramycin. Presumably, at sub-MIC concentrations (e.g. 1 ug/ml), planktonic cells would be able to respond by increasing expression of efflux pumps that would be affected by a triclosan-induced decrease in membrane potential. However, Fig S2 clearly shows no synergy. The explanation in lines 452-454 (the difference in tolerance mechanisms available to biofilm/stationary cells versus planktonic cells) makes a lot more sense.

- Fig 2: In the response, the authors state that letter groups are included. However, they are not in the revised figure. This makes it difficult to evaluate certain conclusions made. For instance, in line 185, the reader cannot assess the partial resistance of the combination relative to the ancestral because the statistics are not shown.

- Lines 203-205: I appreciate the changes the authors made to this section to include a more extensive discussion of the various mutations they saw up front in the Results instead of the Discussion. But I am a bit confused by the concluding sentence in this paragraph. While I agree that the data shows that the evolved isolates are more resistant to tobramycin and less resistant to the combination AND that killing by tobramycin is essential for the combination to be effective, I do not follow the logic of combining these two statements. How is the second suggested by the first? In fact, it appears that the combination has the same killing as triclosan alone in Fig 2 (though perhaps this might be clarified by showing the full pairwise statistical comparisons).

- Lines 324-327: In response to the concern about whether biofilm starvation reduces cellular ATP levels, the authors state that the ATP levels in untreated biofilms after 5 days of starvation is lower than before starvation, citing Fig S8 (which is great and exactly what was asked for). However, this data is not in Fig S8. While it does show similar CFU/ml levels, the BacTiterGlo luminescences of the pre- and post-starvation biofilm are missing.

Reviewer #2: No new concerns.

Reviewer #3: Line 32. I suggest changing, “primarily the result of…” to “can result from…” because the authors also cite literature indicating that adaptive resistance can change the constituents of membranes.

Line 61. Delete “unique and…” because adaptive resistance is not unique to P. aeruginosa.

PLOS authors have the option to publish the peer review history of their article (what does this mean?). If published, this will include your full peer review and any attached files.

Reviewer #1: No

Reviewer #2: No

Reviewer #3: No

---

## [Editor Report · Acceptance letter]

20 Oct 2020

Dear Dr. Waters,

We are delighted to inform you that your manuscript, "Triclosan depletes the membrane potential in *Pseudomonas aeruginosa* biofilms inhibiting aminoglycoside induced adaptive resistance," has been formally accepted for publication in PLOS Pathogens.

Best regards,

Kasturi Haldar

Editor-in-Chief

PLOS Pathogens

orcid.org/0000-0001-5065-158X

Michael Malim

Editor-in-Chief

PLOS Pathogens

orcid.org/0000-0002-7699-2064